# AnyRotate: Gravity-Invariant In-Hand Object Rotation with Sim-to-Real Touch

**Max Yang**[1], **Chenghua Lu**[1], **Alex Church**[2], **Yijiong Lin**[1], **Chris Ford**[1], **Haoran Li**[1],
**Efi Psomopoulou**[1], **David A.W. Barton**[1*], **Nathan F. Lepora**[1*]

[1]University of Bristol    [2]Cambrian Robotics

https://maxyang27896.github.io/anyrotate/

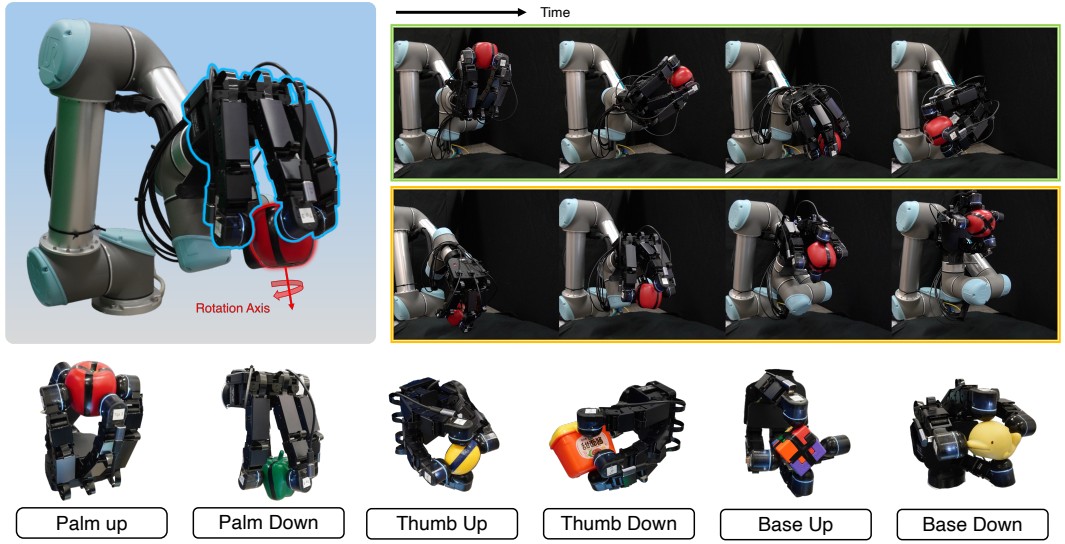

Figure 1: Setup: A 4-fingered 16-DoF tactile robot hand attached to a UR5 performing multi-axis in-hand object rotation (*top*), with experiments in six key hand orientations with respect to gravity: palm up, palm down, thumb up, thumb down, base up and base down (*bottom*).

**Abstract:** Human hands are capable of in-hand manipulation in the presence of different hand motions. For a robot hand, harnessing rich tactile information to achieve this level of dexterity still remains a significant challenge. In this paper, we present AnyRotate, a system for gravity-invariant multi-axis in-hand object rotation using dense featured sim-to-real touch. We tackle this problem by training a dense tactile policy in simulation and present a sim-to-real method for rich tactile sensing to achieve zero-shot policy transfer. Our formulation allows the training of a unified policy to rotate unseen objects about arbitrary rotation axes in any hand direction. In our experiments, we highlight the benefit of capturing detailed contact information when handling objects of varying properties. Interestingly, we found rich multi-fingered tactile sensing can detect unstable grasps and provide a reactive behavior that improves the robustness of the policy.

**Keywords:** Tactile Sensing, In-hand Object Rotation, Reinforcement Learning

## 1 Introduction

The versatility of manipulating objects of varying shapes and sizes has been a long-standing goal for robot manipulation [1]. However, in-hand manipulation with multi-fingered hands can be hugely

---

\* These authors contributed equally.
Correspondence to max.yang@bristol.ac.uk.

8th Conference on Robot Learning (CoRL 2024), Munich, Germany.

challenging due to the high degree of actuation, fine motor control, and large environment uncertainties. While significant advances have been made in recent years, most prominently the work by OpenAI [2, 3], they have relied primarily on vision-based systems which are not necessarily well suited to this task due to significant self-occlusion. Overcoming these issues often requires multiple cameras and complicated setups that are not representative of natural embodiment.

More recently, researchers have begun to explore the object rotation problem with proprioception and touch sensing [4, 5], treating it as a representative task of general in-hand manipulation. The ability to rotate objects around any chosen axis in any hand orientation displays a useful set of primitives for manipulating objects freely in space, even while the hand is in motion. However, this can be challenging as the object is in an intrinsically unstable configuration without any supporting surfaces, as noted in [6, 7], and requires high-precision control of secure grasps in the presence of gravity (*i.e.* gravity invariant): it is harder to hold an object while manipulating it if the palm is not facing upwards. Tactile sensing is expected to play a key role here as it enables the capture of detailed contact information to better control the robot-object interaction. However, rich tactile sensing for in-hand dexterous manipulation has not yet been fully exploited due to the large sim-to-real gap, often leading to a reduction of high-resolution tactile data to low-dimensional representations [8, 9]. One might expect that a more detailed tactile representation could increase in-hand dexterity and enable new tasks.

In this paper, we introduce AnyRotate: a robot system for performing multi-axis gravity-invariant in-hand object rotation with dense featured sim-to-real touch. Here, we propose to tackle this challenge with sim-to-real RL and rich tactile sensing. We first present our goal-conditioned formulation and dense tactile representation to train an accurate and precise policy for multi-axis object rotation. We then train a tactile perception model to simultaneously predict contact pose and contact force readings from tactile images and capture important features for precise manipulation under noisy conditions. In the real world, we mount tactile sensors onto the fingertips of a four-fingered fully-actuated robot hand to provide rich tactile feedback for performing stable in-hand object rotation.

Our principal contributions are, in summary:
1) An RL formulation using auxiliary goals for learning a unified policy to perform in-hand object rotation about any desired axis for any hand orientation relative to gravity.
2) A dense tactile representation, consisting of contact pose and contact force, for learning in-hand manipulation in simulation. We highlight the benefit of these tactile modalities for handling unseen objects with various physical properties, such as mass and shape.
3) An approach to achieve zero-shot sim-to-real tactile policy transfer, validated on 10 diverse objects in the real world. Our rich tactile policy demonstrates strong robustness across various hand directions and rotation axes and maintains high performance when deployed on a rotating hand.

## 2   Related Work

**Classical Control.** Due to the complexity of the contact physics in dexterous manipulation, work on this topic has traditionally relied on simplified models [10–17]. With improved hardware and design, these methods have continued to demonstrate an increased level of dexterity [18–26]. While these methods offer performance guarantees, they are often limited by the underlying assumptions.

**Dexterous Manipulation.** With advances of machine learning, learning-based control has become a popular approach to achieving dexterity [2, 3, 27, 28]. However, most prior works rely on vision as the primary sense for object manipulation [6, 7, 29, 30], which requires continuous tracking of the object in a highly dynamic scene, and occlusions could lead to poorer performance. Vision also has difficulty capturing local contact information which may be crucial for contact-rich tasks.

More recently, researchers have explored the in-hand object rotation task using proprioception and touch sensing [4, 5, 9, 31]. This has so far been limited to rotation about the primary axes or training separate policies for arbitrary rotation axes with an upward facing hand [8, 32]. In-hand manipulation in different hand orientations can be challenging as the hand must perform finger-gaiting while

keeping the object stable against gravity. Several works [7, 33–35] achieved manipulation with a downward-facing hand using either a gravity curriculum or precision grasp manipulation, but the policies were still limited to a single hand orientation. In this work, we make significant advancements to train a unified policy to rotate objects about any chosen rotation axes in any hand direction, and for the first time achieve in-hand manipulation with a continuously moving and rotating hand.

**Tactile Sensing.** Vision-based tactile sensors have become increasingly popular due to their affordability, compactness, and ability to provide precise and detailed spatial information about contact through high-resolution tactile images [36–39]. However, this fine-grained local contact information has not yet been fully utilized for in-hand dexterous manipulation. Previous studies have reduced the high-resolution tactile images to binary contact [9] or discretized contact location [8] to reduce the sim-to-real gap. In contrast, our system utilizes a dense featured tactile representation consisting of the full contact pose and contact force. We show that this tactile representation can capture important interaction physics that is valuable for dexterous manipulation under unknown disturbances.

**Sim-to-real Methods:** Learning in simulation for tactile robotics has gained appeal as it avoids the practical limitations of large data collection in real-world interactions. This trend has been driven by advancements in high-fidelity tactile simulators [40–43] and various sim-to-real approaches [44–48]. Several works have proposed using high-frequency rendering of tactile images for sim-to-real RL [44, 49, 50]. However, this can be computationally expensive and inefficient, limiting these methods to simpler robotic systems. In this work, we extend the sim-to-real framework of Yang et al. [51] by proposing an approach to predict full contact pose and contact force and apply it to a dexterous manipulation task with a robot hand.

## 3   Method

We perform in-hand object rotation via stable precision grasping, constituting a process for continuous object rotation without a supporting surface. Gravity invariance is considered by randomly initializing hand orientations between episodes. This is a difficult exploration problem whereby the movement of the fingers and object has a constant requirement of maintaining stability, as any finger misplacement can induce slip and lead to an irreversible state where the object is dropped. To obtain a general policy for multi-axis in-hand object rotation, we formulate the object-rotation problem as object reorientation and adopt a two-stage learning process. First, the teacher is trained with privileged information and reinforcement learning [52]. We use an auxiliary goal formulation and adaptive curriculum to achieve sample-efficient training. The student is then trained via supervised learning to imitate the teacher's policy given only real-world observations. During both stages, we provide the agent with rich tactile feedback. To bridge the sim-to-real gap for rich tactile sensing, we collect contact data to train a tactile perception model, which allows for zero-shot policy transfer to the real world. An overview of the method is shown in Figure 2 and 3.

### 3.1   Multi-axis In-hand Object Rotation

We formulate the task as a finite horizon goal-conditioned Markov Decision Process (MDP) $\mathcal{M} = (\mathcal{S}, \mathcal{A}, \mathcal{R}, \mathcal{P}, \mathcal{G})$, defined by a continuous state $s \in \mathcal{S}$, a continuous action space $a \in \mathcal{A}$, a probabilistic state transition function $p(s_{t+1}|s_t, a_t) \in \mathcal{P}$, a goal $g \in \mathcal{G}$ and a reward function $r \in \mathcal{R} : \mathcal{S} \times \mathcal{A} \times \mathcal{G} \rightarrow \mathbb{R}$. At each time step $t$, a learning agent selects an action $a_t$ from the current policy $\pi(a_t|s_t, g)$ and receives a reward $r$. The aim is to obtain a policy $\pi_\theta^*$ parameterized by $\theta$ that maximizes the expected return $\mathbb{E}_{\tau \sim p_\pi(\tau), g \sim q(g)} \left[ \sum_{t=0}^{T} \gamma^t r(s_t, a_t, g) \right]$ over an episode $\tau$.

**Observations.** The observation $O_t$ contains the current and target joint position $q_t, \bar{q}_t \in \mathbb{R}^{16}$, previous action $a_{t-1} \in \mathbb{R}^{16}$, fingertip position $f_t^p \in \mathbb{R}^{12}$, fingertip orientation $f_t^r \in \mathbb{R}^{16}$, binary contact $c_t \in \{0, 1\}^4$, contact pose $P_t \in \mathbb{S}^8$, contact force magnitude $F_t \in \mathbb{R}^4$, and the desired rotation axis $\hat{k} \in \mathbb{S}^2$. The privileged information provided to the teacher includes object position, object orientation, object angular velocity, gravity force vector, and the current goal orientation. Full details can be found in Table 5.

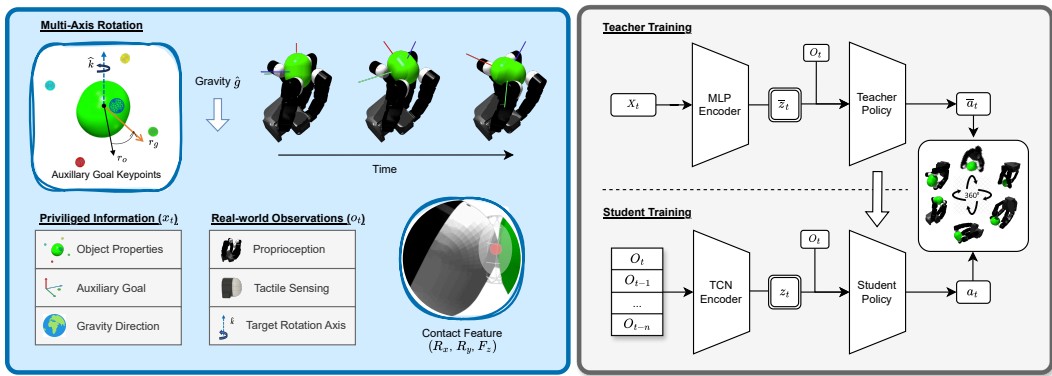

Figure 2: Overview of the approach. *Left*: The object rotation problem is formulated as an object reorientation to a moving goal. Auxiliary goal keypoints are used to define target poses about the chosen rotation axis. *Right*: training a policy using teacher-student policy distillation. The teacher is trained using privileged information with RL and the student aims to imitate the teacher's action given real-world observations. Privileged information and real-world observation are shown.

**Action Space.** At each time step, the action output from the policy is $a_t := \Delta\theta \in \mathbb{R}^{16}$, the relative joint positions of the robot hand. To encourage smooth finger motion, we apply an exponential moving average to compute the target joint positions defined as $\bar{q}_t = \bar{q}_{t-1} + \tilde{a}_t$, where $\tilde{a}_t = \eta a_t + (1 - \eta)a_{t-1}$. We control the hand at $20\,\text{Hz}$ and limit the action to $\Delta\theta \in [-0.026, 0.026]^{16}$ rad.

**Simulated Touch.** We approximate the sensor as a rigid body and fetch the contact information from its sensing surface; the local contact position $(c_x, c_y, c_z)$ for computing contact pose, and the net contact force $(F_x, F_y, F_z)$ for computing contact force magnitude. We apply an exponential moving average on the contact force readings to simulate sensing delay due to elastic deformation. We also saturate and re-scale the contact values to the sensing ranges experienced in reality. Contact force is used to compute binary contact signals using a threshold similar to the real sensor.

**Auxiliary Goal.** When training a unified policy for multi-axis rotation, a formulation using angular velocity can lead to inefficient training and convergence difficulties, as will be shown in Section 5.1. Instead, we formulate the problem as object reorientation to a moving target. Targets are generated by rotating the current object orientation about the desired rotation axis in regular intervals. When a target is reached, a new one is generated about the rotation axis until the episode ends.

**Reward Design.** In the following, we provide an intuitive explanation of the goal-based reward used for learning multi-axis object rotation (with full details in Appendix B):

$$r = r_{\text{rotation}} + r_{\text{contact}} + r_{\text{stable}} + r_{\text{terminate}}. \tag{1}$$

The object rotation objective is defined by $r_{\text{rotation}}$. We use a keypoint formulation $\mathcal{K}(||k_i^{\text{o}} - k_i^{\text{g}}||)$ to define target poses [29] and apply keypoint distance threshold to provide a goal update tolerance $d_{\text{tol}}$. We augment this reward with a sparse bonus reward when a goal is reached and a delta rotation reward to encourage continuous rotation. Next, we use $r_{\text{contact}}$ to maximize contact sensing which rewards tip contacts and penalizes contacts with any other parts of the hand. We also include several terms to encourage stable rotations $r_{\text{stable}}$ comprising: an object angular velocity penalty; a hand-pose penalty on the distance between the joint position from a canonical pose; a controller work-done penalty; and a controller torque penalty. Finally, we include an early termination penalty $r_{\text{terminate}}$, if the object falls out of the grasp or the rotation axis deviates too far from the desired axis.

**Adaptive Curriculum.** The precision-grasp object rotation task can be separated into two key phases of learning: first to stably grasp the object in different hand orientations, then to rotate objects stably about the desired rotation axis. Whilst the $r_{\text{contact}}$ and $r_{\text{stable}}$ reward terms are beneficial for the sim-to-real transfer of the final policy, these terms can hinder the learning process, resulting in local optima where the object will be stably grasped without being rotated. To alleviate this issue, we apply a reward curriculum coefficient $\lambda_{\text{rew}}(r_{\text{contact}} + r_{\text{stable}})$, which increases linearly with the average number of rotations achieved per episode.

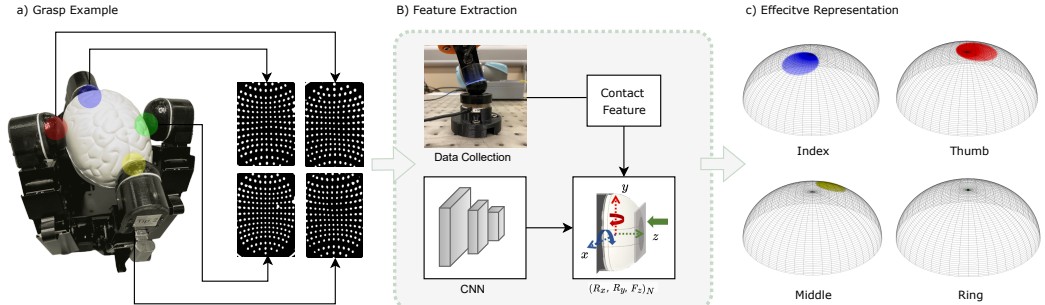

Figure 3: Tactile prediction pipeline; a) tactile images are preprocessed to grey-scale filtered images, b) models extract explicit contact features, c) visualization of the tactile features: contact pose and contact force are represented by the center and area of the shaded circle respectively.

## 3.2 Teacher-Student Policy Distillation

The training in Section 3.1 uses privileged information, such as object properties and auxiliary goal pose. Similar to previous work [5, 8], we use policy distillation to train a student that only relies on proprioception and tactile feedback. The student policy has the same actor-critic architecture as the teacher policy $a_t = \pi_\theta(O_t, a_{t-1}, z_t)$ and returns a Gaussian distribution with diagonal covariances $a_t \equiv \mathcal{N}(\mu_\theta, \Sigma_\theta)$. The latent vector $z_t = \phi(O_t, O_{t-1}, ..., O_{t-n})$ is the predicted low dimensional encoding from a sequence of $N$ proprioceptive and tactile observations. We use a temporal convolutional network (TCN) encoder for the latent vector function $\phi(.)$.

**Training.** The student encoder is randomly initialized and the policy network is initialized with the weights from the teacher policy. We train both the encoder and policy network via supervised learning, minimizing the mean squared error (MSE) of the latent vectors $z_t$ and $\bar{z}_t$ and negative log-likelihood loss (NLL) of the action distributions $a_t$ and $\bar{a}_t$. Without explicit object or goal information, we found the student policy unable to achieve the same level of goal-reaching accuracy as the teacher, which can lead to missing the goal and collecting out-of-distribution data. To alleviate this issue, we increase the goal update tolerance $d_{\text{tol}}$ during student training.

## 3.3 Sim-to-Real Dense Featured Touch

For sim-to-real transfer, we train a tactile perception model to extract contact features from real tactile images [51]. The dense tactile features consist of contact pose and contact force. We use spherical coordinates defined by the contact pose variables: polar angle $R_x$ and azimuthal angle $R_y$. The contact force variable is the magnitude of the 3D contact force $||F||$.

**Data Collection.** We use a 6-DoF UR5 robot arm with the tactile sensor attached to the end effector and a F/T sensor placed on the workspace platform. The tactile sensor is moved on the surface of the flat stimulus at randomly sampled poses. For each interaction, we store tactile images with the corresponding pose and force labels. We then train a CNN model to extract these explicit features of contact from tactile images. More details are given in Appendix H.

**Deployment.** We use the Structured Similarity Index (SSIM) to compute binary contact, which is also used to mask contact pose and force predictions. Given tactile images on each fingertip, we use the tactile perception models to obtain the dense contact features. This is then used as tactile observations for the policy. An overview of the tactile prediction pipeline is shown in Figure 3.

## 4 System Setup

**Real-world.** We use a 16-DoF Allegro Hand with finger-like front-facing vision-based tactile sensors attached to each of its fingertips. Each sensor can be streamed asynchronously along with the joint positions from the hand. The target joint commands are sent with a control rate of 20 Hz. The hand is attached to the end effector of a UR5 to provide different hand orientations for performing in-hand object rotation, as shown in Figure 1.

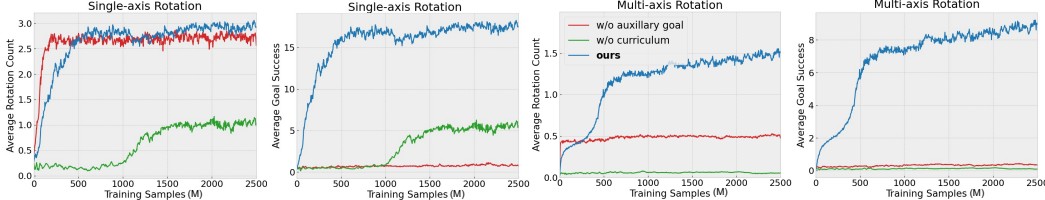

Figure 5: Learning curve for different training strategies. *Left*: Single-axis training fixed about rotation in the z-axis. *Right:* Multiple-axis training for arbitrary rotation axis. We report on the average rotation count and the number of successive goals reached.

**Simulation.** We use IsaacGym [53] for training the teacher and student policies. Each environment contains a simulated Allegro Hand with tactile sensors attached on each fingertip. Gravity is enabled for both the hand and the object. We perform system identification on simulation parameters in various hand directions to reduce the sim-to-real gap (detailed in Appendix D). We run the simulation at $dt = 1/60s$ and policy control at 20 Hz.

**Object Set.** We use fundamental geometric shapes in Isaac Gym (capsule and box) for training. In simulation, we test on two out-of-distribution (OOD) object sets (see Figure 4): 1) OOD Mass, training objects with heavier mass; 2) OOD shape, selection of unseen objects with different shapes. In the real world, we select 10 objects with different properties (see Table 10) to test generalizability of the policy.

**Evaluation** We run each experiment for 600 steps (equating to 30 seconds) and use the following metrics for evaluation:

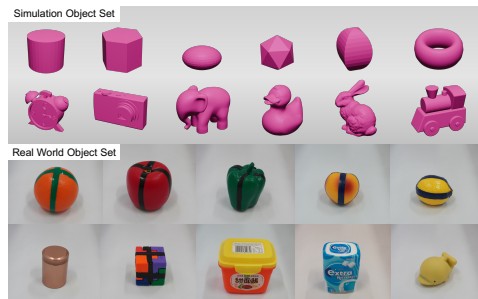

Figure 4: *Top:* Simulation test object set from [54]. *Bottom:* Real everyday objects.

*(i) Rotation Count (Rot)* - the total number of rotations about the desired axis achieved per episode. In the real world, this is manually counted using reference markers attached to the object (visible as the tape in Figure 4).

*(ii) Time to Terminate (TTT)* - time is taken before the object gets stuck, falls out of grasp, or if the rotation axis has deviated away from the target.

## 5 Experiments and Analysis

First we investigate our auxiliary goal formulation and adaptive curriculum for learning the multi-axis object rotation task (Section 5.1). We then study the importance of rich tactile sensing for learning this dexterous manipulation task and conduct a quantitative analysis on the generalizability of the learned polices (Section 5.2). Finally, using the proposed sim-to-real approach, we deploy the policies in the real world on a range of different object rotation tasks (Section 5.3).

### 5.1 Training Performance

We compare our auxiliary goal formulation against angular rotation ("w/o auxiliary goal"), a common formulation for in-hand object rotation [5, 8, 32]. The learning curves are shown in Figure 5. While the agent can learn in the single-axis setting using an angular rotation objective, it resulted in much lower accuracy with near-zero successive goals reached. In the multi-axis setting, the training was unsuccessful and the learning tends to get stuck where the object is stably grasped with minimal rotation. We suspect this is due to the object being held in an intrinsically unstable configuration whereby small random actions can lead to irrecoverable states, such as dropping the object, leading to the agent taking overly conservative actions. During training, when the angular velocity is low and the rotation axis can be noisy, an angular velocity reward cannot effectively guide the agent out of this local optimum. Conversely, provided with privileged goals and a smoother goal-driven reward, the objective becomes more conducive to learning the multi-axis in-hand object rotation task. The proposed adaptive curriculum also contributes positively to this.

## 5.2 Simulation Results

The results for randomized rotation axes in random hand orientations are shown in Table 1. First, we observe that a policy trained in a fixed hand orientation performed poorly in arbitrary hand orientations, suggesting gravity invariance adds considerable complexity to the task. Table 1 compares our dense touch policy (contact pose and contact force) with policies trained with proprioception, binary touch, and discrete touch (a discretized representation introduced in [8]).

| Tactile Observation | OOD Mass | | OOD Shape | |
|---|---|---|---|---|
| | Rot | EpLen(s) | Rot | EpLen(s) |
| Fixed Hand Orn | $0.55_{\pm 0.06}$ | $11.8_{\pm 0.2}$ | $0.55_{\pm 0.04}$ | $19.1_{\pm 0.5}$ |
| Proprioception | $1.34_{\pm 0.07}$ | $21.5_{\pm 0.5}$ | $0.82_{\pm 0.02}$ | $25.1_{\pm 0.3}$ |
| Binary Touch | $1.90_{\pm 0.04}$ | $20.8_{\pm 0.5}$ | $1.57_{\pm 0.05}$ | $25.3_{\pm 0.2}$ |
| Discrete Touch | $1.95_{\pm 0.15}$ | $22.2_{\pm 0.4}$ | $1.67_{\pm 0.08}$ | $26.5_{\pm 0.1}$ |
| Dense Force (w/o Pose) | $2.05_{\pm 0.04}$ | $22.0_{\pm 0.8}$ | $1.60_{\pm 0.02}$ | $25.5_{\pm 0.4}$ |
| Dense Pose (w/o Force) | $2.05_{\pm 0.05}$ | $21.9_{\pm 0.1}$ | $1.73_{\pm 0.03}$ | $26.7_{\pm 0.0}$ |
| **Dense Touch (Ours)** | $\mathbf{2.18_{\pm 0.05}}$ | $\mathbf{22.8_{\pm 0.8}}$ | $\mathbf{1.77_{\pm 0.01}}$ | $\mathbf{27.2_{\pm 0.3}}$ |

Table 1: Comparison of different tactile policies on test object sets in simulation. We report on average rotation achieved per episode (Rot) and average episode length (EpLen) for arbitrary rotation axis and hand direction.

Contrary to the findings in [8], we find binary touch to be beneficial over proprioception alone. We attribute this to including binary contact information during teacher training, which provides a better base policy. Overall, we found that performance improved with more detailed tactile sensing. The dense touch policy, trained with information regarding contact pose and force, outperformed policies that used simpler, less detailed touch. Moreover, discretizing the contact location led to a drop in performance, suggesting that this type of representation is not as well suited to the morphology of our front-facing sensor.

**Ablation Studies.** The results for each tactile modality showed that contact force can provide useful information regarding the interaction physics when handling objects with different mass properties; contact pose is beneficial when handling unseen shapes; and excluding either feature of dense touch resulted in suboptimal performance.

## 5.3 Real-world Results

Object rotation performance for various hand orientations and rotation axes are given in Tables 2 and 3. In both cases, the dense touch policy performed the best, demonstrating a successful transfer of the dense tactile observations. The proprioception and binary touch policies were less effective at maintaining stable rotation, often resulting in loss of contact or getting stuck.

**Hand orientations.** The performance dropped as the hand directions changed from palm up and palm down, followed by base up and base down, to the thumb up and thumb down directions. We attribute this to the larger sim-to-real gap when fingers are positioned horizontally during manipulation. In the latter cases, the gravity loading of the fingers acts against actuation, which weakens the hand in those orientations. However, despite the noisy system, a policy provided with rich tactile information consistently demonstrated more stable performance. Examples are shown in Figure 6.

**Rotation Axis.** Rotation about $z$-axis was the easiest to achieve, followed by the $x$- and $y$-axes. We noticed that binary touch performed less well compared to proprioception when rotating about the $z$-axis, but performed better for $x$- and $y$-rotation axes. The latter axes require two fingers to hold

| Tactile Observation | x-axis | | y-axis | | z-axis | |
|---|---|---|---|---|---|---|
| | Rot | TTT(s) | Rot | TTT(s) | Rot | TTT(s) |
| Proprioception | $0.35_{\pm 0.33}$ | $16.6_{\pm 12.6}$ | $0.17_{\pm 0.19}$ | $8.33_{\pm 8.5}$ | $1.05_{\pm 0.37}$ | $25.3_{\pm 4.0}$ |
| Binary Touch | $0.87_{\pm 0.43}$ | $26.5_{\pm 5.4}$ | $0.25_{\pm 0.18}$ | $15.9_{\pm 10.5}$ | $0.89_{\pm 0.28}$ | $23.8_{\pm 4.6}$ |
| **Dense Touch (Ours)** | $\mathbf{1.33_{\pm 0.50}}$ | $\mathbf{28.6_{\pm 2.8}}$ | $\mathbf{0.79_{\pm 0.37}}$ | $\mathbf{27.8_{\pm 4.8}}$ | $\mathbf{1.33_{\pm 0.44}}$ | $\mathbf{28.2_{\pm 3.1}}$ |

Table 3: Real-world results for different rotation axes in the palm-down configuration. We report on average rotation count and time to terminate (TTT) per episode.

| Tactile Observation | Palm Up | | Palm Down | | Base Up | | Base Down | | Thumb Up | | Thumb Down | |
|---|---|---|---|---|---|---|---|---|---|---|---|---|
| | Rot | TTT(s) | Rot | TTT(s) | Rot | TTT(s) | Rot | TTT(s) | Rot | TTT(s) | Rot | TTT(s) |
| Proprioception | $1.47_{\pm 0.69}$ | $27.6_{\pm 3.8}$ | $1.05_{\pm 0.37}$ | $25.3_{\pm 4.0}$ | $0.84_{\pm 0.30}$ | $26.8_{\pm 3.6}$ | $0.87_{\pm 0.46}$ | $22.8_{\pm 9.6}$ | $0.78_{\pm 0.53}$ | $20.3_{\pm 9.9}$ | $0.51_{\pm 0.65}$ | $9.50_{\pm 8.9}$ |
| Binary Touch | $1.32_{\pm 0.52}$ | $25.5_{\pm 6.5}$ | $0.89_{\pm 0.28}$ | $23.8_{\pm 4.6}$ | $0.86_{\pm 0.32}$ | $25.3_{\pm 6.2}$ | $0.77_{\pm 0.28}$ | $23.0_{\pm 4.7}$ | $0.83_{\pm 0.49}$ | $22.6_{\pm 9.0}$ | $0.47_{\pm 0.32}$ | $13.2_{\pm 5.7}$ |
| **Dense Touch (Ours)** | $\mathbf{1.57_{\pm 0.57}}$ | $\mathbf{30.0_{\pm 0.0}}$ | $\mathbf{1.33_{\pm 0.44}}$ | $\mathbf{28.2_{\pm 3.1}}$ | $\mathbf{1.32_{\pm 0.32}}$ | $\mathbf{29.8_{\pm 0.6}}$ | $\mathbf{1.17_{\pm 0.38}}$ | $\mathbf{29.4_{\pm 1.8}}$ | $\mathbf{1.08_{\pm 0.47}}$ | $\mathbf{27.9_{\pm 3.1}}$ | $\mathbf{0.91_{\pm 0.33}}$ | $\mathbf{29.2_{\pm 2.0}}$ |

Table 2: Real-world results of policies trained on different observations for rotating about the z-axis in different hand directions. We report on average rotation count (Rot) and time to terminate (TTT) per episode over all test objects.

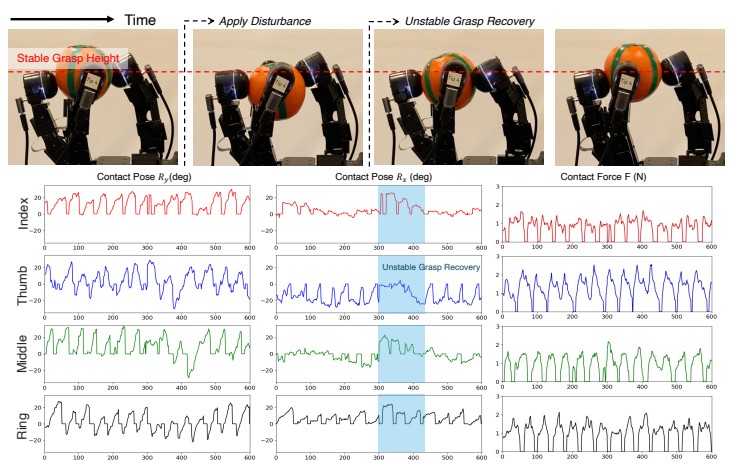

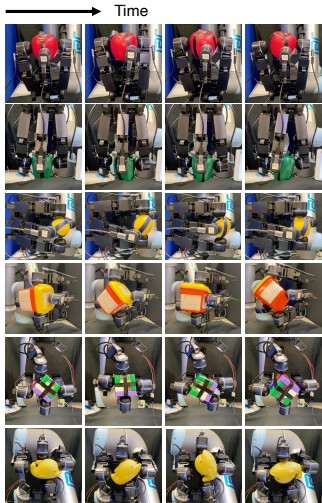

Figure 6: Frames of in-hand object rotation for six distinct objects under six hand orientations relative to gravity.

Figure 7: Tactile features during rollout. The rotation component of contact is seen in $R_y$, a repeated cycle of the object rolling along the fingertips. A reactive behavior is seen in the blue-shaded region in $R_y$, where after boundary contact detection, the fingers extend to reduce contact angle in subsequent cycles to achieve more stable grasping. A demonstration is included on our website.

the object steady (middle/thumb or index/pinky) while the remaining two fingers provide a stable rotating motion. This requires more sophisticated finger-gaiting, and the policy struggled to perform well without tactile sensing.

**Emergent Behavior.** An analysis of the tactile predictions during a perturbed rollout is shown Figure 7. We apply a grasp offset to the object at $n_{\text{step}} = 300$ to visualize the robustness of the policy. The two key motions for stable object rotation can be seen in the output contact pose and force. Given rich tactile sensing on a multi-fingered hand, the policy can detect unstable grasps under boundary contact and provide reactive finger-gaiting motions that prevent the object from slipping further. This emergent behavior was not seen when using proprioception or binary touch.

**Gravity Invariance.** We also demonstrate that the trained policy can adapt effectively to a rotating hand where the gravity vector is continuously changing in the hand's frame of reference. Sample performance for three hand trajectories are provided in the Appendix K.4 and on our website. This capability to manipulate objects during angular movements of the hand enables 6D reorientation of the object while simultaneously repositioning the grasp location. This gives a new level of dexterity for robot hands that could be beneficial in many tasks, *e.g.* general pick-and-place.

## 6 Conclusion and Limitations

In this paper, we demonstrated the capability of a general policy leveraging rich tactile sensing to perform in-hand object rotation about any rotation axis in any hand direction. This marks a significant step toward more general tactile dexterity with fully-actuated multi-fingered robot hands.

We found the policies had difficulties with objects that have sharp geometric features, such as corners or edges, and rotating with these features required a more adaptive policy. To improve the performance further, a richer tactile representation and perception model can be used to better capture these precise geometric features, such as the pose of an edge feature, or incorporate visual information about the object's shape. Also, the actuation of the Allegro Hand was significantly weakened under certain hand orientations. Therefore, designing low-cost and more capable hardware is crucial for advancing dexterous manipulation with multi-fingered robotic hands.

The goal to manipulate objects effortlessly in free space using a sense of touch mirrors a key aspect of human dexterity and stands as a significant goal in robot manipulation. We hope that our research underscores the importance of tactile sensing and spurs continued efforts towards this goal.

**Acknowledgments**

We thank Andrew Stinchcombe for helping with the 3D-printing of the stimuli and tactile sensors. We thank Haozhi Qi for the valuable discussions. This work was supported by the EPSRC Doctoral Training Partnership (DTP) scholarship.

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

# A  Observations and Privileged Information

The full list of real-world observations $o_t$ and privileged information $x_t$ used for the agent is presented in Tables 4 and 5 respectively. The proprioception and tactile dimensions are in multiples of four, representing four fingers.

| Name | Symbol | Dimensions |
|---|---|---|
| **Proprioception** | | |
| Joint Position | $q$ | 16 |
| Fingertip Position | $f^p$ | 12 |
| Fingertip Orientation | $f^o$ | 16 |
| Previous Action | $a_{t-1}$ | 16 |
| Target Joint Positions | $\bar{q}$ | 16 |
| **Tactile** | | |
| Binary Contact | $c$ | 4 |
| Contact Pose | $P$ | 8 |
| Contact Force Magnitude | $F$ | 4 |
| **Task** | | |
| Target Rotation Axis | $\hat{k}$ | 3 |

Table 4: Full list of observations $o_t$ available in simulation and the real world used for teacher and student policy.

| Name | Symbol | Dimensions |
|---|---|---|
| **Object Information** | | |
| Position | $p_o$ | 3 |
| Orientation | $r_o$ | 4 |
| Angular Velocity | $\omega_r$ | 3 |
| Dimensions | $\dim_o$ | 2 |
| Center of Mass | $\mathrm{COM}_o$ | 3 |
| Mass | $m_o$ | 1 |
| Gravity Vector | $\hat{g}$ | 3 |
| **Auxiliary Goal Information** | | |
| Position | $p_g$ | 3 |
| Orientation | $r_g$ | 4 |

Table 5: Full list of privileged information $x_t$ only available in simulation.

The privileged information is used to train the teacher with RL and for obtaining the target latent vector $\bar{z}$ during student training. Whilst the gravity vector can be inferred using the end effector pose of the robot arm, we did not find any benefit of explicitly including this information in the policy. We suspect that using a history of observation, which includes proprioception and contact forces, can also implicitly infer the gravity direction.

# B  Reward Function

## B.1  Base Reward

We use the following reward function for learning multi-axis in-hand object rotation:

$$r = r_{\mathrm{rotation}} + r_{\mathrm{contact}} + r_{\mathrm{stable}} + r_{\mathrm{terminate}}, \tag{2}$$

where,

$$r_{\mathrm{rotation}} = \lambda_{\mathrm{kp}} r_{\mathrm{kp}} + \lambda_{\mathrm{rot}} r_{\mathrm{rot}} + \lambda_{\mathrm{goal}} r_{\mathrm{goal}},$$
$$r_{\mathrm{contact}} = \lambda_{\mathrm{rew}} (\lambda_{\mathrm{gc}} r_{\mathrm{gc}} + \lambda_{\mathrm{bc}} r_{\mathrm{bc}}),$$
$$r_{\mathrm{stable}} = \lambda_{\mathrm{rew}} (\lambda_{\omega} r_{\omega} + \lambda_{\mathrm{pose}} r_{\mathrm{pose}} + \lambda_{\mathrm{work}} r_{\mathrm{work}} + \lambda_{\mathrm{torque}} r_{\mathrm{torque}}),$$
$$r_{\mathrm{terminate}} = \lambda_{\mathrm{penalty}} r_{\mathrm{penalty}}$$

The key terms for defining the object rotation task are $r_{\mathrm{rotation}}$ and $r_{\mathrm{penalty}}$. We include contact terms to encourage fingertip dexterity and tactile sensing. We also include various stability terms commonly used in previous work [8, 32] to obtain natural-looking policies and aid the sim-to-real transfer. In the following, we explicitly define each term of the reward function.

*Keypoint Distance Reward:*

$$r_{\mathrm{kp}} = \frac{d_{\mathrm{kp}}}{(e^{ax} + b + e^{-ax})} \tag{3}$$

where the keypoint distance $kp_{\mathrm{dist}} = \frac{1}{N} \sum_{i=1}^{N} ||k_i^o - k_i^g||$, $k^o$ and $k^g$ are keypoint positions of the object and goal respectively. We use $N = 6$ keypoints placed $5\,\mathrm{cm}$ from the object origin in each of its principle axes, and the parameters $a = 50$, $b = 2.0$.

*Rotation Reward:*

$$r_{\text{rot}} = \text{clip}(\Delta\Theta \cdot \hat{k}; -c_1, c_1) \tag{4}$$

The rotation reward represents the change in object rotation about the target rotation axis. We clip this reward in the limit $c_1 = 0.025$rad.

*Goal Bonus Reward:*

$$r_{\text{goal}} = \begin{cases} 1 & \text{if } kp_{\text{dist}} < d_{\text{tol}} \\ 0 & \text{otherwise} \end{cases} \tag{5}$$

where we use a keypoint distance tolerance $d_{\text{tol}}$ to determine when a goal has been reached.

*Good Contact Reward:*

$$r_{\text{gc}} = \begin{cases} 1 & \text{if } n_{\text{tip\_contact}} \geq 2 \\ 0 & \text{otherwise} \end{cases} \tag{6}$$

where $n_{\text{tip\_contact}} = \text{sum}(c)$. This rewards the agent if the number of tip contacts is greater or equal to 2 to encourage stable grasping contacts.

*Bad Contact Penalty:*

$$r_{\text{bc}} = \begin{cases} 1 & \text{if } n_{\text{non\_tip\_contact}} \geq 0 \\ 0 & \text{otherwise} \end{cases} \tag{7}$$

where $n_{\text{non\_tip\_contact}}$ is defined as the sum of all contacts with the object that is not a fingertip. We accumulate all the contacts in the simulation to calculate this.

*Angular Velocity Penality:*

$$r_\omega = -\min(||\omega_o|| - \omega_{\text{max}}, 0) \tag{8}$$

where the maximum angular velocity $\omega_{\text{max}} = 0.6$. This term penalises the agent if the angular velocity of the object exceeds the maximum.

*Pose Penalty:*

$$r_{\text{pose}} = -||q - q_0|| \tag{9}$$

where $q_0$ is the joint positions for some canonical grasping pose.

*Work Penalty:*

$$r_{\text{work}} = -\tau^T \bar{q} \tag{10}$$

*Torque Penalty:*

$$r_{\text{work}} = -||\tau|| \tag{11}$$

where in the above $\tau$ is the torque applied to the joints during an actioned step.

*Termination Penalty:*

$$r_{\text{terminate}} = \begin{cases} -1 & (k\, p_{\text{dist}} > d_{\text{max}}) \text{ or } (\hat{k}_o > \hat{k}_{\text{max}}) \\ 0 & \text{otherwise} \end{cases} \tag{12}$$

Here we define two conditions to signify the termination of an episode. The first condition represents the object falling out of grasp, for which we use the maximum keypoint distance of $d_{\text{max}} = 0.1$. The second condition represents the deviation of the object rotation axis from the target rotation axis ($\hat{k}_o$) beyond a maximum $\hat{k}_{\text{max}}$. We use $\hat{k}_{\text{max}} = 45°$.

The corresponding weights for each reward term is: $\lambda_{\text{kp}} = 1.0$, $\lambda_{\text{rot}} = 5.0$, $\lambda_{\text{goal}} = 10.0$, $\lambda_{\text{gc}} = 0.1$, $\lambda_{\text{bc}} = 0.2$, $\lambda_\omega = 0.5$, $\lambda_{\text{pose}} = 0.5$, $\lambda_{\text{work}} = 0.1$, $\lambda_{\text{torque}} = 0.05$, $\lambda_{\text{penalty}} = 50.0$.

## B.2  Alternative Reward

We also formulate an alternative reward function consisting of an angular velocity reward and rotation axis penalty to compare with our auxiliary goal formulation.

*Angular Velocity Reward:*

$$r_{\mathrm{av}} = \mathrm{clip}(\omega \cdot \hat{k}, -c_2, c_2) \tag{13}$$

where $c_2 = 0.5$.

*Rotation Axis Penalty:*

$$r_{\mathrm{axis}} = 1 - \frac{\hat{k} \cdot \hat{k}_o}{||\hat{k}||||\hat{k}_o||} \tag{14}$$

where $||\hat{k}_o||$ is the current object rotation axis.

We form the new $r_{\mathrm{rotation}}$ reward $r_{\mathrm{rotation}} = \lambda_{\mathrm{av}} r_{\mathrm{av}} + \lambda_{\mathrm{rot}} r_{\mathrm{rot}}$. We provide an additional object axis penalty $\lambda_{\mathrm{axis}} r_{\mathrm{axis}}$ in the $r_{\mathrm{stable}}$ term and remove the angular velocity penalty, $\lambda_\omega = 0$. The weights are $\lambda_{\mathrm{av}} = 1.5$ and $\lambda_{\mathrm{axis}} = 1.0$. We keep all other terms of the reward function the same.

## B.3  Adaptive Reward Curriculum

The adaptive reward curriculum is implemented using a linear schedule of the reward curriculum coefficient $\lambda_{\mathrm{rew}}(r_{\mathrm{contact}} + r_{\mathrm{stable}})$ which increases with successive goals are reached per episode,

$$\lambda_{\mathrm{rew}} = \frac{g_{\mathrm{eval}} - g_{min}}{g_{max} - g_{min}} \tag{15}$$

where $[g_{min}, g_{max}]$ determines the ranges where the reward curriculum is active. This changes the learning objective towards more realistic finger-gaiting motions as the contact and stability reward increases. We use $[g_{min}, g_{max}] = [1.0, 2.0]$.

## C  Grasp Generation

To generate stable grasps, we initiate the object at 13cm above the base of the hand at random orientations and initialize the hand at a canonical grasp pose at the palm-up hand orientation. We then sample relative offset to the joint positions $\mathcal{U}(-0.3, 0.3)$ rad. We run the simulation by 120 steps (6 seconds) while sequentially changing the gravity direction from 6 principle axes of the hand ($\pm xyz$-axes). We save the object orientation and joint positions (10000 grasp poses per object) if the following conditions are satisfied:
- The number of tip contacts is greater than 2.
- The number of non-tip contacts is zero
- Total fingertip to object distance is less than 0.2
- Object remains stable for the duration of the episode.

## D  System Identification

To reduce the sim-to-real gap of the allegro hand, we perform system identification to match the simulated robot hand with the real hand. We model each of the 16 DoF of the hand with the parameters; stiffness, damping, mass, friction, and armature, resulting in a total of 80 parameters to optimize. We collect corresponding trajectories in simulation and the real world in various hand orientations and use CMA-ES [55] to minimize the mean-squared error of the trajectories to find the best matching simulation parameters.

## E    Domain Randomization

In addition to the initial grasping pose, target rotation axis and hand orientation, we also include additional domain randomization during teacher and student training to improve sim-to-real performance (shown in Table 6).

| Object | | Hand | |
|---|---|---|---|
| Capsule Radius (m) | [0.025, 0.034] | PD Controller: Stiffness | $\times\mathcal{U}(0.9, 1.1)$ |
| Capsule Width (m) | [0.000, 0.012] | PD Controller: Damping | $\times\mathcal{U}(0.9, 1.1)$ |
| Box Width (m) | [0.045, 0.06] | Observation: Joint Noise | 0.03 |
| Box Height (m) | [0.045, 0.06] | Observation: Fingertip Position Noise | 0.005 |
| Mass (kg) | [0.025, 0.20] | Observation: Fingertip Orientation Noise | 0.01 |
| Object: Friction | 10.0 | | |
| Hand: Friction | 10.0 | **Tactile** | |
| Center of Mass (m) | [-0.01, 0.01] | Observation: Pose Noise | 0.0174 |
| Disturbance: Scale | 2.0 | Observation: Force Noise | 0.1 |
| Disturbance: Probability | 0.25 | | |
| Disturbance: Decay | 0.99 | | |

Table 6: Domain randomization parameters.

## F    Simulated Tactile Processing

To simulate our soft tactile sensor in a rigid body simulator, we process the received contact information from the simulator to make up the tactile observations. We use contact force information to compute binary contact signals:

$$c = \{1 \text{ if } ||\mathbf{F}|| > 0.25 \, N; 0 \text{ otherwise}\} \tag{16}$$

A contact force threshold of $0.25 \, N$ was selected to simulate the binary contact detection of the real sensor. For contact force information, we simulate sensing delay caused by elastic deformation of the soft tip in the real world by applying an exponential average on the received force readings.

$$\mathbf{F} = \alpha F_t + (1 - \alpha)F_{t-1} \tag{17}$$

We use $\alpha = 0.5$. We then apply a saturation limit and re-scaling to align simulated contact force sensing ranges with the ranges experienced in the real world.

$$\mathbf{F} = \beta_F \text{clip}(F, \ F_{\min}, \ F_{\max}) \tag{18}$$

We use $\beta_F = 0.6$, $F_{\min} = 0.0 \, N$, $F_{\max} = 5.0 \, N$. We also apply the same saturation and rescaling factor for the contact pose.

$$\mathbf{P} = \beta_P \text{clip}(P, \ P_{\min}, \ P_{\max}) \tag{19}$$

We use $\beta_P = 0.6$, $P_{\min} = -0.53 \, \text{rad}$, $P_{\max} = 0.53 \, \text{rad}$. We use binary contact signals to mask contact pose and contact force observations to minimize noise in the tactile feedback. The same masking technique was applied in the real world.

## G    Architecture and Policy Training

The network architecture and training hyperparameters are shown in Table 7. The proprioception policy uses an observation input dimension of $N = 79$, the binary touch $N = 83$, and the full touch $N = 95$. We use a history of 30 time steps as input to the temporal convolutional network (TCN) and encode the privileged information into a latent vector of size $n = 8$ for all the policies.

| Teacher | | Student | |
|---|---|---|---|
| MLP Input Dim | 18 | TCN Input Dim | [30, N] |
| MLP Hidden Units | [256, 128, 8] | TCN Hidden Units | [N, N] |
| MLP Activation | ReLU | TCN Filters | [N, N, N] |
| Policy Hidden Units | [512, 256, 128] | TCN Kernel | [9, 5, 5] |
| Policy Activation | ELU | TCN Stride | [2, 1, 1] |
| Learning Rate | $5 \times 10^{-3}$ | TCN Activation | ReLU |
| Num Envs | 8192 | Latent Vector Dim $z$ | 8 |
| Rollout Steps | 8 | Policy Hidden Units | [512, 256, 128] |
| Minibatch Size | 32768 | Policy Activation | ELU |
| Num Mini Epochs | 5 | Learning Rate | $3 \times 10^{-4}$ |
| Discount | 0.99 | Num Envs | 8192 |
| GAE $\tau$ | 0.95 | Batch Size | 8192 |
| Advantage Clip $\epsilon$ | 0.2 | Num Mini Epochs | 1 |
| KL Threshold | 0.02 | Optimizer | Adam |
| Gradient Norm | 1.0 | Goal Update $d_{\text{tol}}$ | 0.25 |
| Optimizer | Adam | | |
| Goal Update $d_{\text{tol}}$ | 0.15 | | |

Table 7: Policy training parameters. Please refer to ref. [56] and [57] for a detailed explanation of each hyperparameter.

## H  Tactile Perception Model

**Data Collection.** The setup for tactile feature extraction is shown in Figure 8. We collect data by tapping and shearing the sensor on a flat stimulus fixed onto a force torque sensor and collect six labels for training: contact depth $z$, contact pose in $R_x$, contact pose in $R_y$, and contact forces $F_x$, $F_y$ and $F_z$. In order to capture sufficient contact features needed for the in-hand object rotation task and stay within the contact distribution, we sample the sensor poses with the ranges shown in Table 8. This provides sensing ranges for contact pose between $[-28°, 28°]$ and contact force of up to 5 N, which are the largest ranges we could reasonably consider for this tactile sensor.

**Training.** The architecture and training parameters of the perception model are shown in Table 9. For each fingertip sensor, we collect 3000 images (2400 train and 600 test) and train separate models. The prediction error for one of the sensors is shown in Figure 9. The perception model does not explicitly consider multiple contact points. In practice, we found this to be rare and by assuming a single combined contact, the model was sufficient in producing consistent estimates that did not affect the final performance of the in-hand rotation policy.

## I  Tactile Image Processing

The tactile sensors provide raw RGB images from the camera module. We use an exposure setting of 312.5 and a resolution of $640 \times 480$, providing a frame rate of up to 30 FPS. The images are then postprocessed. We convert the raw image to greyscale and resale the dimension to $240 \times 135$.
**Binary Contact:** We apply a medium blur filter with an aperture linear size of 11, followed by an adaptive threshold with a block size of 55 pixels and a constant offset value of -2. These operations improve the smoothness of the image and filter out noise. The postprocessed image is compared with a reference image using the Structural Similarity Index (SSIM) to compute binary contact (0 or 1). We use an SSIM threshold of 0.6 for contact detection.
**Contact Pose and Force:** We directly use the resized greyscale image for contact force and pose prediction. From the target labels, we use contact pose ($R_x$, $R_y$) and the contact force components $F_x$, $F_y$, $F_z$ (to compute the contact force magnitude $||F||$) to construct the dense tactile representation used during policy training. We use the binary contact signal to mask contact pose and force, thresholding the predictions at $\approx 0.25N$.

| Pose Component | Sampled range |
|---|---|
| Depth $z$ (mm) | [-1, -4] |
| Shear $S_x$ (mm) | [-2, -2] |
| Shear $S_y$ (mm) | [-2, -2] |
| Rotation $R_x$ (deg) | [-28, 28] |
| Rotation $R_y$ (deg) | [-28, 28] |

Table 8: Sensor pose sampling ranges used during tactile data collection for training the pose and force prediction models, relative to the sensor coordinate frame.

| Tactile Perception Model | |
|---|---|
| Conv Input Dim | [240, 135] |
| Conv Filters | [32, 32, 32, 32] |
| Conv Kernel | [11, 9, 7, 5] |
| Conv Stride | [1, 1, 1, 1] |
| Max Pooling Kernal | [2, 2, 2, 2] |
| Max Pooling Stride | [2, 2, 2, 2] |
| Output Dim | 6 |
| Batch Normalization | True |
| Activation | ReLU |
| Learning Rate | $1 \times 10^{-4}$ |
| Batch Size | 16 |
| Num Epochs | 100 |
| Optimizer | Adam |

Table 9: Tactile perception model training parameters.

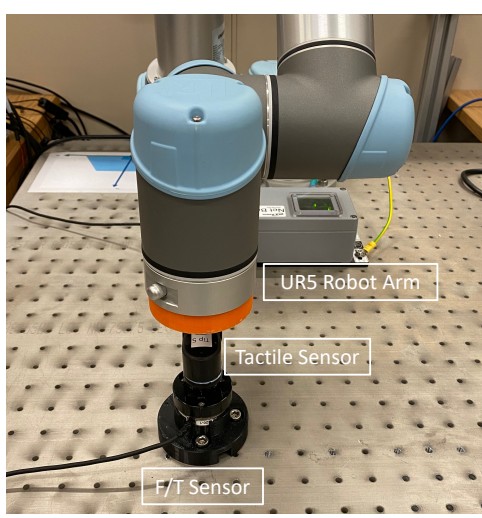

Figure 8: Data collection setup for training tactile perception model, including an F/T sensor and a UR5 Robot arm.

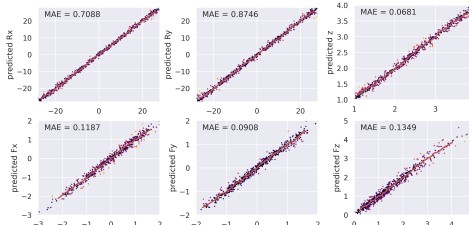

Figure 9: Error plots on test data for the perception model.

## J  Real-world Deployment

**Tactile Sensor Design.** This design of the sensor is based on the DigiTac version [39] of the Tac-Tip [37, 58], a soft optical tactile sensor that provides contact information through marker-tipped pin motion under its sensing surface. Here, we have redesigned the DIGIT base to be more compact with a new PCB board, modular camera and lighting system (Figure 10). We also improved the morphology of the skin and base connector to provide a larger and smoother sensing surface for greater fingertip dexterity. The tactile sensor skin and base are entirely 3D printed with Agilus 30 for skin and vero-series for the markers on the pin-tips and for the casings. Each base contains a camera

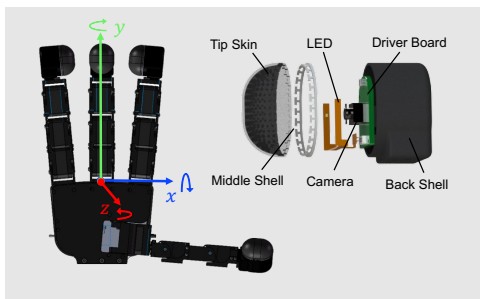

Figure 10: CAD models of the fully-actuated (Allegro) robot hand and integrated custom tactile sensors.

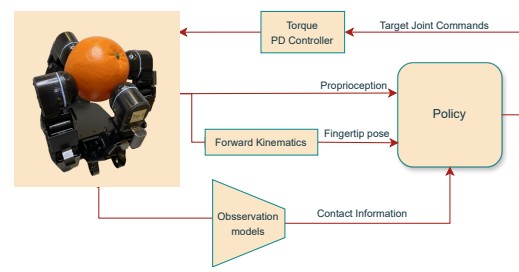

Figure 11: Real-world robot hand control pipeline.

|  | **Real-world Object Set** | | | | |
|---|---|---|---|---|---|
|  | Dimensions (mm) | Mass (g) |  | Dimensions (mm) | Mass (g) |
| Apple | $75 \times 75 \times 70$ | 60 | Tin Cylinder | $45 \times 45 \times 63$ | 30 |
| Orange | $70 \times 72 \times 72$ | 52 | Cube | $51 \times 51 \times 51$ | 65 |
| Pepper | $61 \times 68 \times 65$ | 10 | Gum Box | $90 \times 80 \times 76$ | 89 |
| Peach | $62 \times 56 \times 55$ | 30 | Container | $90 \times 80 \times 76$ | 32 |
| Lemon | $52 \times 52 \times 65$ | 33 | Rubber Toy | $80 \times 53 \times 48$ | 27 |

Table 10: Dimensions and mass of real-world everyday objects.

driver board that connects to the computer via a USB cable and can be streamed asynchronously at a frame rate of 30 FPS. We perform post-processing using OpenCV [59] in real-time.

**Sensor Placement.** A common limitation of unidirectional tactile sensors is that they are primarily sensorized over a front-facing area. Contacts with the side of the sensor casing can be slippery and result in unstable behaviors. To alleviate this issue, similar to [31], we adjusted the sensor direction relative to the fingers to maximize contact with the sensing surface, and placed the tactile fingertips with offsets (thumb, index, middle, ring) = $(-45°, -45°, 0°, 45°)$. This allowed the policies to achieve consistent and stable in-hand object rotation performance, providing a basis to validate our learning approach against baselines.

**Control Pipeline.** Each tactile perception model is deployed together with the policy as shown in Figure 11. We stream tactile and proprioception readings asynchronously at 20 Hz. The joint positions are used by a forward kinematic solver to compute fingertip position and orientation. The relative joint positions obtained from the policy are converted to target joint commands. This is published to the Allegro Hand and converted to torque commands by a PD controller at 300 Hz.

**Object Properties.** Various physical properties of the objects used in the real-world experiment are shown in Table 10. We include objects of different sizes and shapes not seen during training.

## K  Additional Experiments

### K.1  Hyperparamters

We provide additional ablation studies to analyze the design choices for our axillary goal formulation. The effect of goal update tolerance $d_{\text{tol}}$ for the student training and the auxiliary goal increment intervals are shown in Table 11.

The performance can be significantly affected by the goal-update tolerance. As the tolerance reduced during student training, the number of average rotations and successive goals reached per episode also reduced. This suggests that the performance of the teacher policy was poorly transferred and the student could not learn the multi-axis object rotation skill effectively. Increasing the goal increment intervals also resulted in fewer rotations achieved.

| **Goal Update Tolerance** | Rot | TTT(s) | #Success | **Goal Increment** | Rot | TTT(s) | #Success |
|---|---|---|---|---|---|---|---|
| $d_{\text{tol}} = 0.15$ | 0.75 | **28.1** | 3.07 | $\theta = 30°$ | **1.77** | **27.2** | **5.26** |
| $d_{\text{tol}} = 0.20$ | 1.36 | 27.7 | 4.48 | $\theta = 40°$ | 1.50 | 26.7 | 4.36 |
| $\mathbf{d_{\text{tol}} = 0.25}$ | **1.77** | 27.2 | **5.26** | $\theta = 50°$ | 1.30 | 27.1 | 3.86 |

Table 11: We compare the performance of policies trained with different design choices in the auxiliary goal formulation. We compare goal update tolerance and goal increment intervals and provide metrics for average successive goals reached, rotation count (Rot), and time to terminate (TTT).

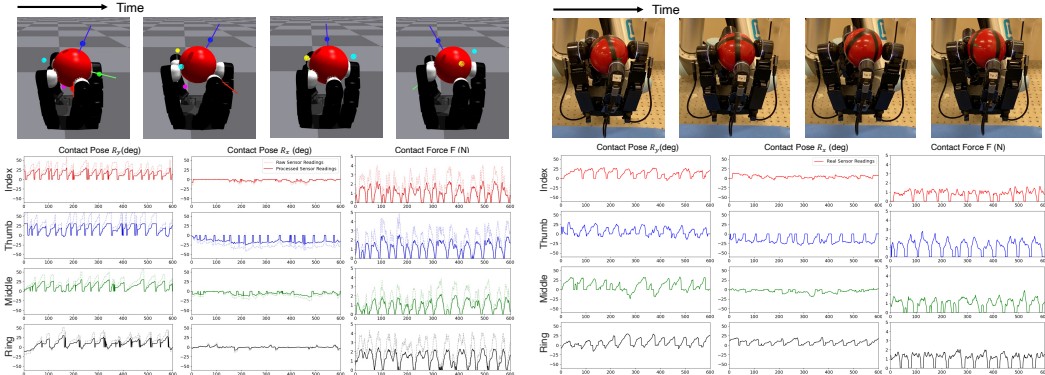

Figure 12: Simulated tactile readings during policy rollout. We plot raw and processed contact pose and contact force readings in simulation for rotating a ball in the palm-up orientation about the z-axis.

Figure 13: Real tactile readings during policy rollout. We plot the contact pose and contact force predictions from the tactile perception model for rotating a ball in the palm-up orientation about the z-axis.

## K.2 Sim-to-Real Tactile Sensing

We present the tactile readings of a similar policy rollout in simulation and real-world, in Figure 12 and 13 respectively. We observed a sim-to-real gap in the rotation speed as demonstrated by the higher contact cycles obtained in simulation. However, a matching pattern can be seen from the recorded contact features. By comparing the raw and processed contact readings in simulation, we see the effect of the post-processing functions in Section F. This helped with aligning the simulated tactile readings to that of the real sensors and smoothing out the noisy readings of the contact force. The comparison also demonstrates a successful sim-to-real transfer of the proposed tactile representation.

## K.3 Real-world Object Results

The real-world results for each object for varying rotation axes and hand orientations are shown in Figure 13 and 12 respectively. We observed that larger objects resulted in fewer rotations, likely due to the size and joint limits of the Allegro Hand, making smaller objects easier to maneuver. Objects with sharp corners, such as the cube and gum box, sometimes caused the fingers to get stuck around these points. We believe that this is because navigating around these geometric features requires additional finger extension during rotation, making it challenging for a general tactile policy to handle such shapes effectively. The gum box was the most challenging due to its sharp corners and shifting mass (sloshing gum pieces). These factors led to the least stable rotation and the lowest time to terminate (TTT).

**Contact Surfaces.** While we train the tactile perception models on a flat surface, we found that it can generalize to other uneven contact surfaces, shown by various test object shapes in Figure 4, demonstrating the robustness of the proposed tactile representation.

## K.4 Rotating Hand

We test the robustness of the policy by performing in-hand object rotation during different hand movements. In particular, we choose hand trajectories where the gravity vector is continuously changing relative to the orientation of the hand, adding greater complexity to this task. Rollouts for three different hand trajectories are shown in Figure 14. In particular, for the third hand trajectory (iii), we demonstrate the capability of the robot hand to servo around the surface of the object in different directions while keeping the object almost stationary in free space. This motion also demonstrates the ability to command different target rotation axes during deployment, offering a useful set of primitives for other downstream tasks.

| Tactile Observation | Apple | | Orange | | Pepper | | Peach | | Lemon | |
|---|---|---|---|---|---|---|---|---|---|---|
| | Rot | TTT(s) | Rot | TTT(s) | Rot | TTT(s) | Rot | TTT(s) | Rot | TTT(s) |
| Proprioception | $0.98_{\pm 0.46}$ | $25.5_{\pm 10.1}$ | $1.21_{\pm 0.51}$ | $25.3_{\pm 7.9}$ | $\mathbf{1.51_{\pm 0.33}}$ | $28.8_{\pm 2.2}$ | $1.54_{\pm 0.42}$ | $26.8_{\pm 2.5}$ | $1.11_{\pm 0.62}$ | $20.3_{\pm 8.5}$ |
| Binary Touch | $1.21_{\pm 0.30}$ | $27.7_{\pm 4.4}$ | $1.26_{\pm 0.47}$ | $26.2_{\pm 6.6}$ | $1.25_{\pm 0.35}$ | $24.3_{\pm 6.8}$ | $1.06_{\pm 0.40}$ | $23.2_{\pm 4.6}$ | $0.86_{\pm 0.54}$ | $19.5_{\pm 7.8}$ |
| **Dense Touch (Ours)** | $\mathbf{1.37_{\pm 0.33}}$ | $\mathbf{30.0_{\pm 0.0}}$ | $\mathbf{1.54_{\pm 0.38}}$ | $\mathbf{30.0_{\pm 0.0}}$ | $1.50_{\pm 0.20}$ | $\mathbf{30.0_{\pm 0.0}}$ | $\mathbf{1.89_{\pm 0.26}}$ | $\mathbf{30.0_{\pm 0.0}}$ | $\mathbf{1.57_{\pm 0.32}}$ | $\mathbf{29.5_{\pm 1.1}}$ |

| Tactile Observation | Tin Cylinder | | Cube | | Gum Box | | Container | | Rubber Toy | |
|---|---|---|---|---|---|---|---|---|---|---|
| | Rot | TTT(s) | Rot | TTT(s) | Rot | TTT(s) | Rot | TTT(s) | Rot | TTT(s) |
| Proprioception | $0.48_{\pm 0.34}$ | $17.0_{\pm 11.5}$ | $0.80_{\pm 0.56}$ | $19.0_{\pm 12.0}$ | $0.57_{\pm 0.46}$ | $19.2_{\pm 10.5}$ | $0.36_{\pm 0.26}$ | $17.7_{\pm 11.3}$ | $0.49_{\pm 0.31}$ | $17.7_{\pm 6.8}$ |
| Binary Touch | $0.44_{\pm 0.20}$ | $16.8_{\pm 8.8}$ | $0.58_{\pm 0.30}$ | $16.8_{\pm 9.1}$ | $0.48_{\pm 0.34}$ | $13.8_{\pm 7.5}$ | $\mathbf{0.59_{\pm 0.21}}$ | $\mathbf{28.7_{\pm 3.0}}$ | $0.65_{\pm 0.25}$ | $21.0_{\pm 7.0}$ |
| **Dense Touch (Ours)** | $\mathbf{0.81_{\pm 0.24}}$ | $\mathbf{28.3_{\pm 3.3}}$ | $\mathbf{0.88_{\pm 0.48}}$ | $\mathbf{23.0_{\pm 10.9}}$ | $\mathbf{0.83_{\pm 0.45}}$ | $\mathbf{24.2_{\pm 11.0}}$ | $\mathbf{0.59_{\pm 0.19}}$ | $27.8_{\pm 3.1}$ | $\mathbf{1.06_{\pm 0.24}}$ | $\mathbf{28.3_{\pm 2.1}}$ |

Table 12: **Hand orientation**. Real-world results of policies trained on different tactile observations for different objects. We report on average rotation count (Rot) and time to terminate (TTT) per episode averaged over the 6 test hand orientations.

| Tactile Observation | Apple | | Orange | | Pepper | | Peach | | Lemon | |
|---|---|---|---|---|---|---|---|---|---|---|
| | Rot | TTT(s) | Rot | TTT(s) | Rot | TTT(s) | Rot | TTT(s) | Rot | TTT(s) |
| Proprioception | $0.53_{\pm 0.26}$ | $23.3_{\pm 9.4}$ | $0.68_{\pm 0.45}$ | $21.7_{\pm 9.7}$ | $0.49_{\pm 0.61}$ | $14.7_{\pm 11.0}$ | $1.00_{\pm 0.41}$ | $28.7_{\pm 1.9}$ | $0.68_{\pm 0.41}$ | $18.7_{\pm 9.0}$ |
| Binary Touch | $0.78_{\pm 0.43}$ | $28.3_{\pm 2.4}$ | $0.90_{\pm 0.57}$ | $23.0_{\pm 9.9}$ | $0.78_{\pm 0.38}$ | $25.3_{\pm 3.3}$ | $0.92_{\pm 0.42}$ | $25.0_{\pm 4.1}$ | $0.88_{\pm 0.38}$ | $20.7_{\pm 7.4}$ |
| **Dense Touch (Ours)** | $\mathbf{1.03_{\pm 0.34}}$ | $\mathbf{30.0_{\pm 0.0}}$ | $\mathbf{1.20_{\pm 0.50}}$ | $\mathbf{30.0_{\pm 0.0}}$ | $\mathbf{1.17_{\pm 0.31}}$ | $\mathbf{30.0_{\pm 0.0}}$ | $\mathbf{1.82_{\pm 0.41}}$ | $\mathbf{30.0_{\pm 0.0}}$ | $\mathbf{1.70_{\pm 0.39}}$ | $\mathbf{30.0_{\pm 0.0}}$ |

| Tactile Observation | Tin Cylinder | | Cube | | Gum Box | | Container | | Rubber Toy | |
|---|---|---|---|---|---|---|---|---|---|---|
| | Rot | TTT(s) | Rot | TTT(s) | Rot | TTT(s) | Rot | TTT(s) | Rot | TTT(s) |
| Proprioception | $0.28_{\pm 0.25}$ | $10.0_{\pm 8.2}$ | $0.63_{\pm 0.45}$ | $20.3_{\pm 10.3}$ | $0.47_{\pm 0.66}$ | $7.33_{\pm 10.4}$ | $0.10_{\pm 0.14}$ | $6.00_{\pm 8.5}$ | $0.35_{\pm 0.38}$ | $14.0_{\pm 12.8}$ |
| Binary Touch | $0.49_{\pm 0.20}$ | $19.3_{\pm 2.9}$ | $0.77_{\pm 0.26}$ | $\mathbf{26.7_{\pm 4.7}}$ | $0.42_{\pm 0.4}$ | $14.7_{\pm 10.7}$ | $0.21_{\pm 0.21}$ | $16.7_{\pm 12.5}$ | $0.47_{\pm 0.41}$ | $15.7_{\pm 15.0}$ |
| **Dense Touch (Ours)** | $\mathbf{0.92_{\pm 0.42}}$ | $\mathbf{29.7_{\pm 0.5}}$ | $\mathbf{1.04_{\pm 0.21}}$ | $24.0_{\pm 4.3}$ | $\mathbf{0.65_{\pm 0.51}}$ | $18.3_{\pm 13.1}$ | $\mathbf{0.42_{\pm 0.12}}$ | $\mathbf{25.0_{\pm 7.1}}$ | $\mathbf{1.29_{\pm 0.19}}$ | $\mathbf{25.7_{\pm 2.1}}$ |

Table 13: **Rotation-axis.** Real-world results of policies trained on different tactile observations for different objects. We report on average rotation count (Rot) and time to terminate (TTT) per episode averaged over the 3 test rotation axes.

i) Hand rotation z-axis

ii) Hand rotation x-axis

iii) Hand rotation about object pose

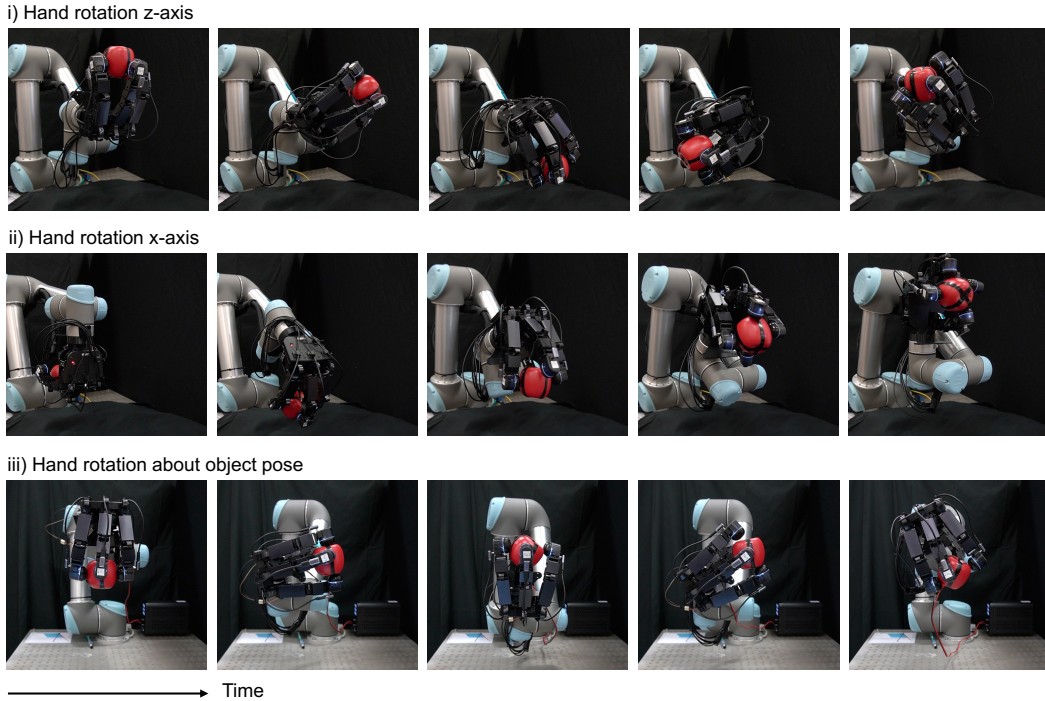

Time

Figure 14: Examples of in-hand object rotation for plastic apple on a moving hand. Rollouts for three hand trajectories are shown: (i) object rotation about $z$-axis while the hand rotates about the $z$-axis from 0 to $2\pi$; (ii) object rotation about z-axis while the hand rotates about the $x$-axis from $-\pi$ to $\pi$; (iii) object rotation about the $y$-axis while the hand rotates about the $y$-axis in the opposite direction to keep the object pose stationary.

