# OpenReview forum: "AnyRotate: Gravity-Invariant In-Hand Object Rotation with Sim-to-Real Touch"
_robot-learning.org/CoRL/2024/Conference — CoRL 2024_

### Official Review · Reviewer_9rrD · 2024-07-21

**Originality:** 2
**Technical Quality:** 3
**Clarity Of Presentation:** 3
**Potential Impact:** 2
**Recommendation:** 3
**Confidence:** 4

**Review:**

The overall quality and clarity of the paper are good. However, the task and method are similar to previous work, which may limit its contribution. While the addition of tactile information and gravity invariance is noteworthy, these changes may not significantly differentiate it from earlier papers focused on axis rotation. Nonetheless, the system engineering aspect, particularly the use of multi-finger tactile sensing, could be considered a major contribution.

# Strength:

1. The paper demonstrates gravity-invariant multi-axis in-hand rotation, with impressive video evidence of object rotation despite changing end-effector orientations.

2. Achieving sim-to-real transfer for tactile-based in-hand reorientation is a significant engineering accomplishment.

3. The paper deals with challenging tactile simulation and sim-to-real transfer.

4. Using a learning method to extract tactile features from images, rather than relying on raw data, is a notable strength.

# Limitation:
See Questions for Rebuttal.

**Quality Of The Limitations Section:**

2

**Questions For Rebuttal:**

# Limitation:

Minor Limitation:

1. The method described in the paper is similar to the approach used in "Visual Dexterity: In-Hand Dexterous Manipulation from Depth" for teaching the pipeline. The main difference is that the authors of the current paper incorporate tactile modalities in the training process.
2. For the hardware design, it would be beneficial if the authors explained the rationale behind choosing different orientations for the tactile sensors on various fingers. Additionally, conducting experiments to compare the benefits of having sensors oriented in different directions versus having all sensors aligned in the same direction would provide valuable insight into the impact of sensor orientation on the system’s performance

Major Limitation:

1. While the dense tactile sensor is commendable at the hardware level, its installation in the hand reveals a potential issue. The authors manually adjusted the sensor direction to face the object, thereby increasing the contact area between the objects and the sensor (as mentioned in Appendix J, lines 587-590). Although this design choice is intuitive and aimed at simplifying the task, it limits the system's generalizability. By optimizing the sensor orientation for this specific task, the hand's versatility for other tasks, even similar rotation tasks with different motions, may be compromised. Therefore, this approach may not serve as a general framework for in-hand rotation tasks. I agree that the digit sensor might not have any signal on the sides of cases, which may be the major reason behind this design. However, I still believe the framework could be more generalizable if the authors had placed all the sensors in the same direction and successfully completed the task.
2. The tactile sensing simulation in the paper raises some questions. The authors treat the sensor as a rigid body to fetch contact information in the simulation. For example, in reality, the contact position will move when the force increases, but this cannot happen in a simulation using a rigid body model. Additionally, it would be beneficial if the authors described how they simulate the delay in contact force response.
3. I assume the authors used Isaac Gym's contact simulation to fetch the contact information without using any external package. It would be great if the authors demonstrated how to obtain contact information in Isaac Gym and visualized both the raw contact data and the processed data. Obtaining accurate contact information in Isaac Gym is challenging, particularly because the contact position can be inaccurate. The study would be much improved if the authors visualized similar contact scenes in both simulation and the real world to compare the tactile representations.
4. In lines 11-13, the authors state, "we found rich multi-fingered tactile sensing can implicitly detect object movement within grasp and provide a reactive behavior that improves the robustness of the policy." This claim needs experimental verification to demonstrate how object movement within the grasp is implicitly detected and how this detection leads to reactive behavior.
5. The task presented is similar to those in previous papers listed in the references, such as "In-Hand Object Rotation via Rapid Motor Adaptation." It would be more impactful if the paper demonstrated enhanced dexterous skills through the use of tactile modalities rather than focusing solely on achieving gravity-invariance. For gravity-invariance, the authors use a relatively simple technique of randomizing initial hand orientations between episodes, which may not be considered a major contribution. It would be beneficial if the authors highlighted how tactile information specifically aids the task, particularly given their significant effort in tactile sim-to-real transfer.
6. The simulation uses a single contact point for tactile information (as visualized in Figure 3), whereas the real-world tactile representation involves a contact area. Bridging the sim-to-real gap between these differing sensing patterns is a key challenge. The paper should address how this discrepancy is managed and how the simulation results translate effectively to the real-world tactile data.

**Robotics Focus:**

4

**Summary Of Paper:**

The paper presents a system designed for gravity-invariant multi-axis in-hand object rotation using a tactile robot hand with rich tactile feedback. The system is trained in simulation and employs zero-shot policy transfer to handle real-world tasks without additional retraining.  Key contributions include:  1. An RL framework for learning a unified policy for multi-axis in-hand object rotation. 2. A dense tactile representation that enhances the handling of objects with different properties. 3. Successful zero-shot sim-to-real policy transfer validated on diverse objects.

**Summary Of Recommendation:**

Weak accept. The paper has contribution on the system engineering with both hardware and learning contribution, but lack of major contribution. I recommend weak reject the paper, but may adjust the score if the issues are well addressed.

---

### Official Review · Reviewer_Si75 · 2024-07-21
**In-hand object rotation with dense tactile sensing: novel method with comprehensive evaluation**

**Originality:** 4
**Technical Quality:** 4
**Clarity Of Presentation:** 4
**Potential Impact:** 3
**Recommendation:** 4
**Confidence:** 3

**Review:**

Overall, the paper is structured and readable. The proposed system is novel and relevant to the CoRL audience. The relevant work seems adequate. The proposed system is clearly presented with sufficient detail for reproductions. The problem formulation and policy training method are generalizable to other robotics tasks. The tactile prediction pipeline can also be applied to various scenarios that require tactile feedback.

Strengths:
- Utilizing contact information provided by dense tactile sensing seems to be a promising approach to dexterous manipulation tasks.
- The zero-shot unified policy transfer demonstrates the robustness of the system in real-world tasks.
- The experiments are thorough, covering various hand orientations, rotation axes, and object properties, with proper baselines selected.
- The ablation experiments indicate that all components of dense tactile sensing are crucial to the performance of the system.

Weaknesses:
- The results are presented by taking the average of results from all tested objects. It is a little unclear how object properties affect the performance. It is possible that the baselines may have better performance than the proposed method in some cases.

Minor comments:
- In Table 3, sign missing at y-axis Dense Touch TTT.

**Quality Of The Limitations Section:**

2

**Questions For Rebuttal:**

1. As the results are presented by taking the average of all tested objects, are there any cases where the baselines outperform the proposed method?
2. The policy is trained using Isaac gym fundamental geometric shapes including boxes, yet the system still has difficulties with box-shaped objects. “Grasping points producing similar tactile information for different states” is a reasonable cause, but is it feasible to replicate this problem in simulation by tuning the simulated touch reading and get a better trained policy?
3. The figures can be improved. As the hand is all black and the fingers look almost the same, it would be helpful to annotate the axes in the figures to help readers understand why rotation axes affect the performance.

**Robotics Focus:**

4

**Summary Of Paper:**

The authors propose a system that leverages dense tactile sensing and reinforcement learning to perform object rotation about arbitrary axes in any hand orientation relative to gravity. They first train a policy using teacher-student policy distillation given rich tactile feedback in simulation, then utilize an observation model trained on contact data to perform zero-shot policy transfer to the real world. They evaluate the system using a 4-fingered 16-DoF tactile robot hand in both simulation and real-world experiments, based on rotations per episode and rotation time until failure.

**Summary Of Recommendation:**

The authors present a system that is relevant and novel, they provide detailed information that helps the readers to understand the system and potentially reproduce it. The system is well grounded and thoroughly tested in both simulation and real-world experiments. The authors might further improve this paper by giving some insights on what external factors affect the system performance. Overall, this is a good paper and I suggest acceptance.

---

### Official Review · Reviewer_mNc5 · 2024-07-22
**Interesting Paper and Good Results, Need to Clarify Several Details**

**Originality:** 3
**Technical Quality:** 3
**Clarity Of Presentation:** 4
**Potential Impact:** 3
**Recommendation:** 3
**Confidence:** 5

**Review:**

Strength:

This paper further pushes the boundaries of in-hand manipulation using sim-to-real by demonstrating how tactile sensing can help the policy maintain stability during changes in the direction of gravity.

This result is particularly interesting as the authors show the usage and analyze the effect of fine-grained touch sensing in this process. This could motivate future researchers in this direction.

The paper is well-written and provides a very comprehensive system design, which can be helpful for the community.

Weakness:

Overall, this paper does a good job of explaining the system design. However, I think this paper could significantly benefit from adding more “insights and discussions” on different choices and experiments. More specifically:
* About the auxiliary goal: one interesting contribution of this paper is the revised reward definition. The current experiment section only describes its better performance compared to the velocity reward. However, from my perspective, the reward is only an integral of the velocity reward and it is not easy to understand why this formulation is better. It would be more helpful to discuss the underlying reason or observations for this design.
* Another interesting observation is the “reactive behavior” mentioned in the abstract. This is discussed in Figure 7. However, I do not see a clear difference between the highlighted blue region and other parts. A more detailed illustration of this would be helpful.

The way this paper collects touch data can only deal with single point contact. If there are multiple points in contact with the finger, or if the contact regions are larger than those used during the data collection time, the force prediction will have problems.

**Quality Of The Limitations Section:**

3

**Questions For Rebuttal:**

Why is the key point-based auxiliary reward better than the rotation velocity reward?

How should we perceive or understand the reactive behavior from Figure 7? For example, what is the unique behavior for the shaded curve, and are the corresponding videos are in the supplementary material?

How important is the gravity direction as an input? In the student training, the gravity direction can also be inferred from the arm’s pose. Have you considered adding this into student policy?

How different are these tactile sensors? Do you need to train a force prediction model for each sensor? Or how do you calibrate between them. In addition, what about the durability of the sensors? How long can the force prediction model be used?

For the dense touch feature data collection, how large can the contact region be? What happens when there are multiple points of contact?

The reward function has quite a few terms. Could the authors comment on the usefulness of them? Which terms are the most critical?

**Robotics Focus:**

4

**Summary Of Paper:**

This paper presents a pipeline for sim-to-real in-hand object manipulation when the hand orientation is randomized. The authors train a tactile perception model that can map tactile images to contact pose and forces. Using this representation, the authors train a policy in simulation and transfer it to the real-world.

**Summary Of Recommendation:**

Overall, I think this is a well-written paper with good results. There are several questions that need to be clarified, but I still recommend acceptance because of the results.

---

### Author Rebuttal · Authors · 2024-08-13

Please see the attached zip file for the rebuttal submission containing a revised manuscript with highlighted changes and an additional supplementary video.

---

### Decision · Program_Chairs · 2024-09-04

**Decision:**

Accept

**Comment:**

The paper makes good advance in the topic of in-hand manipulation with tactile input and achieved good results. The authors should address the clarification questions posted by the reviewers.
---
During the rebuttal phase, the authors improved the paper and addressed most concerns raised by the reviewers. The reviewers all agree to accept the paper now.